# A Novel Sandwich ELASA Based on Aptamer for Detection of Largemouth Bass Virus (LMBV)

**DOI:** 10.3390/v14050945

**Published:** 2022-04-30

**Authors:** Xinyue Zhang, Zemiao Zhang, Junrong Li, Xiaohong Huang, Jingguang Wei, Jiahui Yang, Lingfeng Guan, Xiaozhi Wen, Shaowen Wang, Qiwei Qin

**Affiliations:** 1College of Marine Sciences, South China Agricultural University, Guangdong Laboratory for Lingnan Modern Agriculture, Guangzhou 510642, China; zhy632739212@126.com (X.Z.); 17878762116@163.com (Z.Z.); 18476728230@163.com (J.L.); huangxh@scau.edu.cn (X.H.); weijg@scau.edu.cn (J.W.); yjhaijy@gmail.com (J.Y.); glf18932100876@126.com (L.G.); wenxiaozhi2022@163.com (X.W.); 2Laboratory for Marine Biology and Biotechnology, Qingdao National Laboratory for Marine Science and Technology, Qingdao 266000, China

**Keywords:** ELASA, aptamer, largemouth bass virus, rapid detection

## Abstract

Largemouth bass virus (LMBV) is a major viral pathogen in largemouth bass culture, usually causing high mortality and heavy economic losses. Accurate and early detection of LMBV is crucial for diagnosis and control of the diseases caused by LMBV. Previously, we selected the specific aptamers, LA38 and LA13, targeting LMBV by systematic evolution of ligands by exponential enrichment (SELEX). In this study, we further generated truncated LA38 and LA13 (named as LA38s and LA13s) with high specificity and affinities and developed an aptamer-based sandwich enzyme-linked apta-sorbent assay (ELASA) for LMBV diagnosis. The sandwich ELASA showed high specificity and sensitivity for the LMBV detection, without cross reaction with other viruses. The detection limit of the ELASA was as low as 1.25 × 10^2^ LMBV-infected cells, and the incubation time of the lysate and biotin labeled aptamer was as short as 10 min. The ELASA could still detect LMBV infection in spleen lysates at dilutions of 1/25, with good consistency of qRT-PCR. For the fish samples collected from the field, the sensitivity of ELASA was 13.3% less than PCR, but the ELASA was much more convenient and less time consuming. The procedure of ELASA mainly requires washing and incubation, with completion in approximately 4 h. The sandwich ELASA offers a useful tool to rapidly detect LMBV rapidly, contributing to control and prevention of LMBV infection.

## 1. Introduction

Largemouth bass (*Micropterus salmoides*) is an important economic freshwater fish, which is widely farmed because of its fast growth, strong adaptability and delicious taste [1,2]. The production of largemouth bass has reached 0.4778 million tons in China in 2019 [3]. However, largemouth bass usually suffers from disease problems, especially caused by viruses such as largemouth bass virus (LMBV), which belongs to the genus Ranavirus, family Iridoviridae and has been recognized as one of the major pathogens in largemouth bass worldwide [4,5]. Typically, LMBV infection causes lethargy, external and internal hemorrhages, and organomegaly, resulting in high mortality and serious economic losses [6,7,8]. Hence, LMBV seriously threatens healthy development in the largemouth bass industry.

Undoubtedly, the early and rapid diagnosis of LMBV is critical for controlling LMBV infection. Presently, PCRs are commonly used methods to detect LMBV, including conventional PCR, qRT-PCR and LAMP [5]. These methods are sensitive but complex procedures, while expensive equipment and high environmental requirement limit their application in largemouth bass aquaculture [9,10]. Therefore, less time consuming, less expensive, and detection environment independent detection methods or diagnosis of LMBV in fields are urgently needed.

In recent years, the rapid detection methods based on aptamer have been widely applied. Aptamers are single-stranded strands of oligonucleotides (DNA or RNA) that are able to bind specifically to targets by folding into secondary or tertiary structures [11,12]. Aptamers are usually obtained by systematic evolution of ligands by exponential enrichment (SELEX) and the targets can be a wide range of proteins, viruses, bacteria, parasites or cells [13,14,15]. Moreover, aptamers pose such excellent characteristics such as easy synthesis in vitro, high stability, low cost, non-toxicity, and flexible modification, that aptamers are recognized as ideal molecular probes and are widely used in diagnosis [16,17,18,19].

The enzyme-linked immunosorbent assay (ELISA) is one of the most classical diagnostic methods, which does not require a complex procedure and special environment. However, antibody-based ELISA has several disadvantages. For example, specific antibodies are difficult to prepare, the procedure is time consuming, and the results are unstable. Therefore, an enzyme-linked apta-sorbent assay (ELASA) was designed, in which aptamers replace antibodies.

We previously generated novel aptamers (LA38 and LA13) targeting LMBV. In this study, truncated LA38 and LA13 (named as LA38s and LA13s) were generated, and their specificity and affinity were identified. The dissociation constants (*K_d_*) of LA38s and LA13s were 3.42 nM and 2.34 nM, respectively. Then, we developed and characterized a novel sandwich ELASA based on LA38s and LA13s for the rapid detection for LMBV. This technology is sensitive and convenient, showing great potential utility for rapid diagnosis of LMBV.

## 2. Methods and Materials

### 2.1. Cultured Cells, Fish and Viruses

Epithelioma papulosum cyprini (EPC) cells were provided by Dr. Wang Qing (Pearl River Fisheries Research Institute, Chinese Academy of Aquatic Sciences, Guangzhou, China). Grouper spleen (GS) cells were established and maintained in our laboratory. EPC cells and GS cells were cultured in Leibovitz’s L-15 medium (Thermo Fisher Scientific, Waltham, MA, USA) containing 10% fetal bovine serum (Thermo Fisher Scientific, Waltham, MA, USA) at 28 °C.

Juvenile largemouth bass (6–8 cm) were purchased from a fishery in Guangdong Province and raised in a circulating water system at 25–30 °C. The fish were fed twice daily for 10 days before they were used. In addition, the fish were randomly selected before the experiment and tested negative for LMBV by PCR.

LMBV was isolated from diseased largemouth bass and stored in our laboratory. Singapore grouper iridovirus (SGIV) was isolated from diseased grouper and preserved in our laboratory. Frog virus 3 (FV3) was kindly gifted by Prof. Wang Xiaolong (Northeast Forestry University, Harbin, China). LMBV or FV3 can proliferate in EPC cells, and SGIV is cultivated in GS cells [4,20,21]. LMBV and SGIV were purified by sucrose gradient ultracentrifugation, as previously described [22]. The purified virus was stored at −80 °C until use.

### 2.2. Aptamers

The aptamer, LA38 and LA13, were selected and characterized in our previous study [22]. In this study, LA38 and LA13 were truncated by removing the terminal sequences used for previous selection. The truncated aptamers were named as LA38s and LA13s. The sequence of LA38s was 5′-GCCGGCCCGGGGGATAGAGTGCTCCCGATCCCTTGGCGAAGGGAC-3′. The sequence of LA13s was 5′-TTTTGACGCTTTATCCTTTTCTTATGGCGGG ATAGTTTCG-3′. The biotin-labelled aptamers were synthesized by Sangon Biotech.

### 2.3. Specificity Analysis of LA38s and LA13s

The 96-well microplate plates were precoated with 50 μL of LMBV at 4 °C for 4 h, and were washed three times with PBS. The aptamers were heated at 95 °C for 5 min and cooled by ice for 10 min for denaturation. Biotin-labeled random initial ssDNA library was used as a control. Subsequently, pre-denatured biotin-aptamers (200 nM) were added to each precoated well at 4 °C for 1 h. After washing with PBS, 100 μL streptavidin-labelled horseradish peroxidase (HRP; 1:10,000) (Thermo Fisher Scientific, Waltham, MA, USA) was added to the wells for 30 min. The wells were carefully washed five times, and were treated by the Pierce TMB Substrate Kit (Thermo Fisher Scientific, Waltham, MA, USA), following the manufacturer’s instructions.

### 2.4. Affinity Analysis of LA38s and LA13s

The 5′-biotinylated aptamers were diluted to various concentrations (0–500 nM), and then added to the LMBV or SGIV precoated wells. SGIV served as the control. Subsequently, the wells were measured using an ELASA, as previously described. After subtracting the mean absorbance at 450 nm (OD_450_) values of the control groups, the apparent equilibrium dissociation constant (*K_d_*) of the aptamer-LMBV interaction was calculated according to the equation: Y = Bmax × X/(*K_d_* + X), using SigmaPlot software v13.0.3.0 [23].

### 2.5. Assembly of Sandwich ELASA

The sandwich ELASA protocol was designed by previously reported assays with some modifications [24,25]. LA38s and LA13s were used to capture and detect targets, respectively, to construct sandwich ELASA model (aptamer-LMBV-aptamer). The biotin-labeled library was used as a control. Briefly, LA38s (200 nmol) was desaturated at 95 °C for 10 min, cooled on ice for 10 min, and then was immobilized on the Pierce^TM^ Streptavidin Coated 96-well Plates to capture targets for 1 h at 4 °C. After incubation, each well was washed three times with 200 μL PBS. Then, 200 μL of D-Biotin was added and incubated for 30 min at 4 °C. Wells were washed three times with 200 μL PBS and added to 50 μL LMBV for 1 h at 4 °C. After washing with PBS, LA13s was added to the wells and incubated at 4 °C for 1 h. Subsequently, the wells were washed, and then 100 µL of horseradish peroxidase labeled with streptavidin was added. Each well was then carefully washed five times and treated by the Pierce TMB Substrate Kit (Thermo Fisher Scientific, Waltham, MA, USA). After termination of the reaction by 2 M H_2_SO_4_, the absorbance of each analyte on the 96-well plate was measured at 450 nm using the microplate reader.

### 2.6. Specificity Analysis of Sandwich ELASA

Ten sandwich models were prepared for specificity analysis, including aptamer (LA38s)-LMBV-aptamer (LA13s) sandwich model, aptamer (LA38s)-lysate of LMBV-infected EPC-aptamer (LA13s) sandwich model, aptamer (LA38s)-SGIV-aptamer (LA13s) sandwich model, aptamer (LA38s)-FV3-aptamer (LA13s) sandwich model, aptamer (LA38s)-EPC-aptamer (LA13s) sandwich model, library-LMBV-library sandwich model, library-lysate of LMBV-infected EPC-library sandwich model, library-SGIV-library sandwich model, library FV3-library sandwich model, library-EPC-library sandwich model.

### 2.7. The Optimum Working Concentration of Aptamer in Sandwich ELASA

Briefly, LMBV (50 μL) (10^6^ TCID_50_/mL) was added to the LA38s precoated wells at 4 °C for 1 h. SGIV was used as the control. After the wells were washed three times with PBS, various concentrations (0–1000 nM) of LA13s were added to the wells and incubated for 1 h at 4 °C. Then the wells were treated by HRP and the Pierce TMB Substrate Kit.

### 2.8. Sensitivity Analysis of Sandwich ELASA

Briefly, LMBV-infected EPC cells were diluted to various concentrations: 9 × 10^7^, 1 × 10^7^, 1 × 10^6^, 1 × 10^5^, 5 × 10^4^, 2.5 × 10^3^, 1.25 × 10^2^. Normal EPC cells and SGIV-infected GS cells were used as controls. The cells were collected, lysed and tested for the sandwich ELASA as described above.

Moreover, the incubation time of the cell lysates and the aptamer was changed from 60 min to 5 min to investigate the influence of the incubation time.

### 2.9. Detection of LMBV in Infected Fish Tissues by Sandwich ELASA

To test the feasibility of ELASA, six groups of fish (*n* = 30) were infected with LMBV by intraperitoneal injection. After five days post injection, the spleen tissues were collected and stored at −80 °C until use. To confirm the LMBV infection of the fish samples, the spleen tissues were tested by qRT-PCR. The specific primers were listed in Table 1. Spleen samples of uninjected largemouth bass were used as controls. RNA was extracted using the SV Total RNA Isolation Kit (Promega, Madison, WI, USA) and used for reverse transcription by the ReverTra Ace qPCR RT Kit (Toyobo, Tokyo, Japan), according to the following reaction procedure. The qRT-PCR were performed using 2× SYBR Green Real-time PCR Mix kit (Toyobo, Tokyo, Japan). Meanwhile, the spleen tissues were lysed by Pierce^TM^ IP Lysis Buffer (Thermo Fisher Scientific, Waltham, MA, USA), and tested by the sandwich ELASA as described above. In addition, spleen sample lysates were diluted to 1/12.5, 1/25, 1/50 and 1/100, and also used for the sandwich ELASA.

Furthermore, 30 largemouth bass suspected of being infected with LMBV were collected from a farm in Guangdong Province. The spleens were collected and divided into two parts for qRT-PCR and sandwich ELASA separately, as mentioned above.

## 3. Results

### 3.1. Characterization of the Truncated Aptamers

The shorter aptamers, LA38s and LA13s, for targeting LMBV were truncated from the reported aptamers, LA38 and LA13, respectively [22]. The specificity and affinity of LA38s and LA13s were further detected. As shown in Figure 1, LA38s and LA13s were both specifically bound to LMBV, rather than SGIV, FV3 and EPC cells. Moreover, LA38s and LA13s showed high affinity for LMBV, with a calculated *K_d_* of 3.42 nM and 2.34 nM, respectively (Figure 2). These results suggested that LA38s and LA13s could target LMBV with high specificity and affinity.

### 3.2. The Specificity of Sandwich ELASA

Compared with SGIV, FV3 and EPC cells, purified LMBV and LMBV-infected cell lysates could be specifically detected by this sandwich ELASA and could specifically detect purified LMBV and LMBV-infected cell lysates. Moreover, the sandwich ELASA based on the random initial ssDNA library, the library-LMBV-library sandwich, could not detect LMBV or LMBV-infected EPC cell (Figure 3). These results demonstrated the specificity of this sandwich ELASA for the detection of LMBV.

### 3.3. Optimum Working Concentration of Aptamers Used in the Sandwich ELASA

Various concentrations of LA13s (0, 25, 50, 100, 200, 250, 500 and 1000 nM) were used to construct the sandwich ELASA. As shown in Figure 4, the sandwich ELASA dose-dependently detected LMBV. The effective detection was observed when the aptamer concentration was as low as 200 nM. When the aptamer was less than 200 nm, the OD_450_ was significantly decreased. However, the OD_450_ in the control group (SGIV used as control target) showed no obvious binding at different concentrations of aptamer. Therefore, it is inferred that the optimal aptamer working concentration for the sandwich ELASA is 200 nM.

### 3.4. Sensitivity Analysis of the Sandwich ELASA

The sensitivity of sandwich ELASA was verified by various concentrations of LMBV-infected EPC cells. As shown in Figure 5A,B, the sandwich ELASA could detect as low as 1.25 × 10^2^ cells/mL LMBV infected cells lysates. Furthermore, this sandwich ELASA could detect LMBV-infected cells even when the incubation time was as short as 10 min (Figure 5C). Therefore, our data showed the high sensitivity of this sandwich ELASA for LMBV detection.

### 3.5. Detection of LMBV in Largemouth Bass by the Sandwich ELASA

As shown in Figure 6A, the qRT-PCR assay of the LMBV MCP gene demonstrated that largemouth bass (group 1 to 6) were infected with LMBV. Obviously, the sandwich ELASA could detect LMBV of the samples, group 1 to 6, even when the sample lysates were diluted to original 1/25.

The existence of LMBV in 30 largemouth bass collected from farms was detected by both qRT-PCR method and sandwich ELASA method. The results of qRT-PCR method showed that 18 (60%) samples were positive of LMBV and 12 (40%) samples were negative of LMBV, and the results of sandwich ELASA method showed that 14 (46.7%) samples were positive of LMBV, and 16 (53.3%) samples were negative of LMBV, meaning that 4 (13.3%) samples were tested by qRT-PCR to be positive but by sandwich ELASA negative. Hence, the detection results of sandwich ELASA showed comparative consistency with that of qRT-PCR, with slightly less sensitive (13.3% lower) (Table 2).

## 4. Discussion

Viral disease outbreaks are quick, so rapid diagnosis is of great significance for prevention and control. In 1995, with the successive deaths of about 1000 adult largemouth bass in South Carolina’s Santee-Cooper, LMBV was identified as the putative etiological agent [26]. After that, LMBV has usually been detected in diseased largemouth bass. The mortality rate of largemouth bass by artificial injection of LMBV reached 100% [27]. In the early stage, LMBV infection generally has no pathognomonic lesions or obvious symptoms on the surface for largemouth bass [7,28]. Hence, it is difficult to judge LMBV infection based on clinical signs. At present, there are four main methods of pathogen detection in clinic: (1) Electron microscopic observation of LMBV. (2) Virus isolation based on cell culture. (3) Detection of specific viral gene based on PCR, qRT-PCR and LAMP. (4) ELISA based on viral protein antibody [4,29,30,31]. Although these detection methods are sensitive, their application in the field is limited, due to the complex and time-consuming operation, high cost and the requirement of sophisticated equipment. Therefore, we have developed a novel sandwich ELASA for the rapid diagnosis of LMBV infection.

The specific aptamers targeting LMBV provide a reliable basis for construction of sandwich ELASA to detect LMBV. In this study, the reported aptamers were truncated, without affecting their specificity and affinity.

The sandwich ELASA includes different models, such as aptamer-target-capture antibody-detect antibody, aptamer-target-capture antibody-aptamer, aptamer-target-antibody, and antibody-target-aptamer [32,33,34]. In this study, the aptamer-target-aptamer model was chosen. In addition, some variations of the term ‘‘ELASA’’ have been developed, such as enzyme-linked oligonucleotide assay (ELONA) and aptamer-linked immobilized sorbent assay (ALISA) [34]. For example, apple stem pitting virus (ASPV) could be detected by ELONA [22].

Here, the sandwich ELASA was sufficiently sensitive to detect LMBV as low as 1.25 × 10^2^/mL LMBV-infected cell lysates. Notability, when the cell lysates was more than 1 × 10^5^/mL, we could judge LMBV infection with naked eyes based on the different color of each lysate, suggesting that the sandwich ELASA could be a simpler detection method without special equipment. Furthermore, when the incubation time was as short as 10 min, the sandwich ELASA could still detect LMBV as low as 1.25 × 10^2^/mL LMBV-infected cell lysates.

Finally, we tested the sensitivity and effectiveness of the sandwich ELASA with fish samples. For artificially infected largemouth bass, the sandwich ELASA could detect LMBV infection in spleen tissues at dilutions of 1/25. For largemouth bass collected from farms, the results of the sandwich ELASA for LMBV detection were consistent with the PCR results, with 13.3% reduction in sensitivity. The PCR is a widely used and highly sensitive assay, but requires complex operation and considerable time, such as DNA or RNA extraction, and electrophoresis detection. In contrast, the whole process of sandwich ELASA mainly includes incubation and washing, and needs ~4 h, suggesting that the sandwich ELASA is a useful tool for LMBV detection in practice.

## 5. Conclusions

In conclusion, we successfully developed a sandwich ELASA for the rapid detection of LMBV infection with high specificity and sensitivity. This method has a wide application prospect in diagnosis in aquaculture.

## Figures and Tables

**Figure 1 viruses-14-00945-f001:**
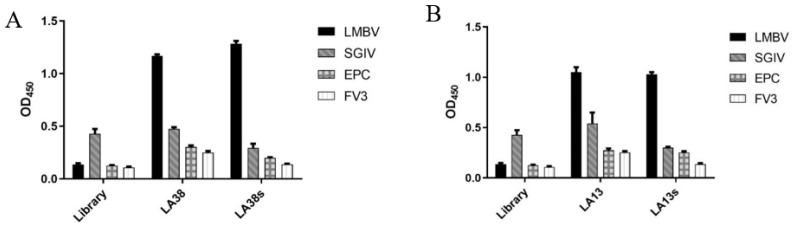
Specificity of truncated aptamers: (**A**) The specificity of the truncated aptamers LA38s and original LA38 against LMBV. (**B**) The specificity of the truncated aptamers LA13s and original LA13 against LMBV. The initial ssDNA Library, SGIV, EPC cells and FV3 served as controls.

**Figure 2 viruses-14-00945-f002:**
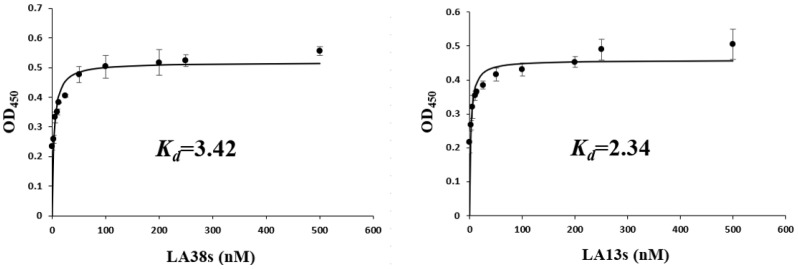
Affinity analysis of truncated aptamers against LMBV. LMBV was incubated with different concentrations (0–500 nM) of 5′-biotinylated aptamers. The specific *K_d_* was calculated using standardized nonlinear regression analysis by SigmaPlot software. Results were presented as the mean ± SD.

**Figure 3 viruses-14-00945-f003:**
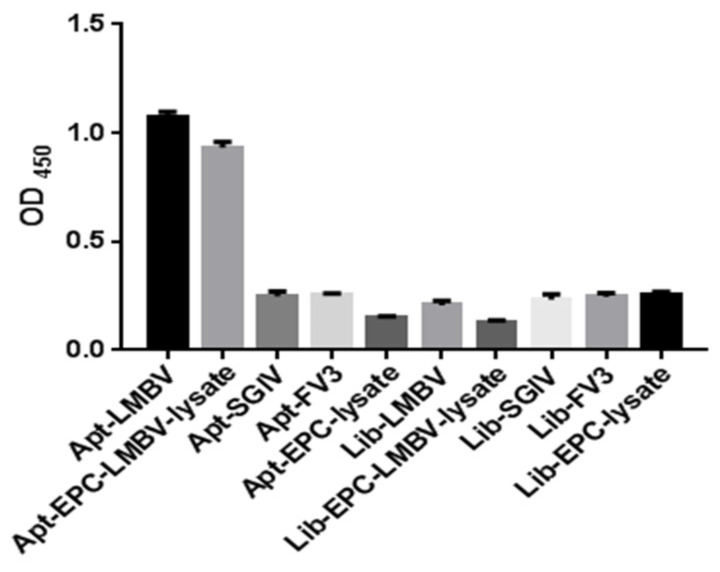
Specificity of the sandwich ELASA. The names along the *X*-axis are illustrated in detail as follows: Apt-LMBV indicates LA38s-LMBV-LA13s sandwich model. Apt-EPC-LMBV-lysate indicates LA38s-the lysates of LMBV-infected EPC-LA13s sandwich model. Apt-SGIV indicates LA38s-SGIV-LA13s sandwich model. Apt-FV3 indicates LA38s-FV3-LA13s sandwich model. Apt-EPC-lysate indicates LA38s-EPC-LA13s sandwich model. Lib-LMBV indicates Library-LMBV-Library sandwich model. Lib-EPC-LMBV-lysate indicates Library-the lysates of LMBV-infected EPC. Lib-SGIV indicates Library-SGIV-Library sandwich model. Lib-FV3 indicates Library-FV3-Library sandwich model. Lib-EPC-lysate indicates Library-EPC-Library sandwich model. SGIV, FV3, EPC and Library were used as control.

**Figure 4 viruses-14-00945-f004:**
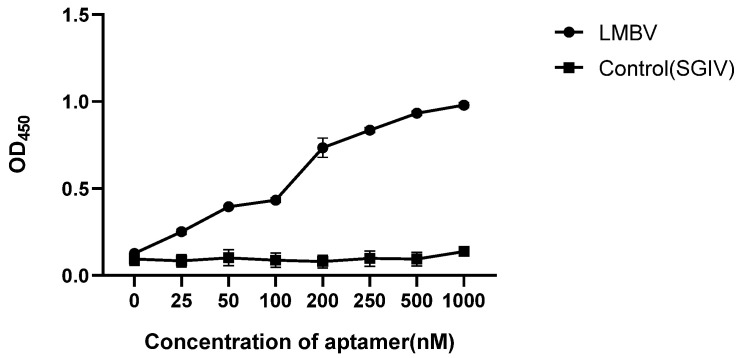
Optimum working concentration of aptamer. The final OD_450_ of the sandwich ELASA was detected with different concentrations of aptamers. SGIV was used as controls.

**Figure 5 viruses-14-00945-f005:**
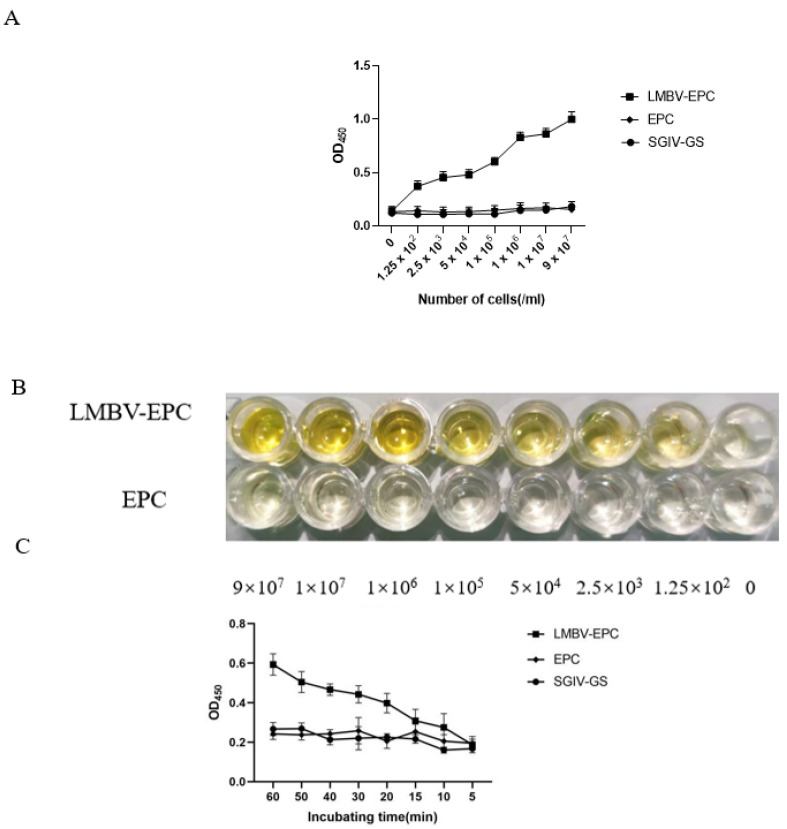
The sensitivity analysis of the sandwich ELASA: (**A**) The sandwich ELASA could detect as low as 1.25 × 10^2^ cells/mL LMBV infected cells lysates. The SGIV-infected GS cells and uninfected EPC cells were used as the control. (**B**) Chromogenic reactions of LMBV-infected EPC lysates. (**C**) The sandwich ELASA detected LMBV infection with an incubation time as short as 10 min. The SGIV-infected GS cells and uninfected EPC cells as the controls.

**Figure 6 viruses-14-00945-f006:**
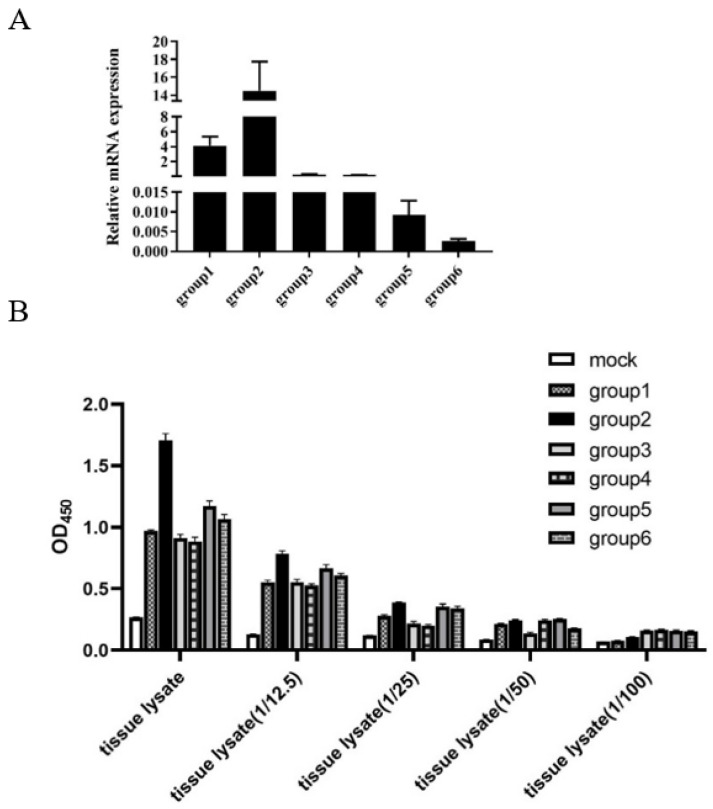
Detection of LMBV in fish samples by sandwich ELASA: (**A**) The results of qRT-PCR for LMBV infection in fish samples. The mean values and SD of three independent experiments were used for analysis. (**B**) The sandwich ELASA detect LMBV infection in samples of spleen lysates even diluted 1/25.

**Table 1 viruses-14-00945-t001:** Primer sequences used in this study.

Primer	Sequence (5′–3′)
LMBV-MCP-qRT-PCR-F	TCGCCACTTATGACAGCCTTGA
LMBV-MCP-qRT-PCR-R	GACCTGGGCACTCCTACGGA
Actin-qRT-PCR-F	TACGAGCTGCCTGACGGACA
Actin-qRT-PCR-R	GGCTGTGATCTCCTTCTGCA

**Table 2 viruses-14-00945-t002:** Sensitivity comparison of sandwich ELASA and qRT-PCR for detection of 30 fishes.

Tested Samples	Positive Samples		Positive Ratios (%)	
	ELASA	qRT-PCR	ELASA	qRT-PCR
30	14	18	46.7	60

## Data Availability

The data that support the findings of this study are available from the corresponding author upon reasonable request.

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
