# Peer review of "A Novel Sandwich ELASA Based on Aptamer for Detection of Largemouth Bass Virus (LMBV)"

_viruses, 2022, doi:10.3390/v14050945_

Round 1

Reviewer 1 Report

English changes are required. For example,

Page 1: “Largemouth bass (Micropterus salmoides), one important economic  freshwater fish, …”

Page 1: “… has been recognized as a fatal pathogen in largemouth bass in worldwide”

Page 8: "30 largemouth bass collected from farms were detected by qRT-PCR and sandwich ELASA respectively. 14 (46.7%) were tested positive by sandwich ELASA, while 18 (60%) were tested positive by qRT-PCR. Only 4 (13.3%) samples showed positive of LMBV in-fection by PCR tested, but showed negative by sandwich ELASA. 12 samples were both tested negative by qRT-PCR and sandwich ELASA. Hence, the detection results of sand-wich ELASA showed good consistency with that of qRT-PCR, with slightly less sensitive (13.3 % lower) (Table 2)."

Page 6: “As shown in Figure 5A&B, the sandwich ELASA could detect as low as 1.25×102 TCID50 or cells?/mL LMBV infected cells lysates”.

Author Response

Response to Reviewer 1 Comments

Point 1: English changes are required. For example,

Page 1: “Largemouth bass (Micropterus salmoides), one important economic freshwater fish, …”

Page 1: “… has been recognized as a fatal pathogen in largemouth bass in worldwide”

Page 8: "30 largemouth bass collected from farms were detected by qRT-PCR and sandwich ELASA respectively. 14 (46.7%) were tested positive by sandwich ELASA, while 18 (60%) were tested positive by qRT-PCR. Only 4 (13.3%) samples showed positive of LMBV in-fection by PCR tested, but showed negative by sandwich ELASA. 12 samples were both tested negative by qRT-PCR and sandwich ELASA. Hence, the detection results of sandwich ELASA showed good consistency with that of qRT-PCR, with slightly less sensitive (13.3 % lower) (Table 2)."

Page 6: “As shown in Figure 5A&B, the sandwich ELASA could detect as low as 1.25×102 TCID50 or cells?/mL LMBV infected cells lysates”.

Response 1: Thanks for the reviewer’s comments. We have revised the manuscript carefully. For example,

Page 1: “Largemouth bass (Micropterus salmoides), one important economic freshwater fish, …” has been revised to “Largemouth bass (Micropterus salmoides) is an important economic freshwater fish, which is widely farmed because of its fast growth, strong adaptability and delicious taste”. 

Page 1: “… has been recognized as a fatal pathogen in largemouth bass in worldwide” has been revised to “However, largemouth bass usually suffered from disease problems, especially caused by viruses such as largemouth bass virus (LMBV), which belongs to the genus Ranavirus, family Iridoviridae and has been recognized as one of the major pathogens in largemouth bass worldwide”.

Page 10: “30 largemouth bass collected from farms were detected by qRT-PCR and sandwich ELASA respectively. 14 (46.7%) were tested positive by sandwich ELASA, while 18 (60%) were tested positive by qRT-PCR. Only 4 (13.3%) samples showed positive of LMBV in-fection by PCR tested, but showed negative by sandwich ELASA. 12 samples were both tested negative by qRT-PCR and sandwich ELASA. Hence, the detection results of sandwich ELASA showed good consistency with that of qRT-PCR, with slightly less sensitive (13.3 % lower) (Table 2)” has been revised to “The existence of LMBV in 30 largemouth bass collected from farms was detected by both qRT-PCR method and sandwich ELASA method. The results of qRT-PCR method showed that 18 (60%) samples were positive of LMBV, and 12 (40%) samples were negative of LMBV, and the results of sandwich ELASA method showed that 14 (46.7%) samples were positive of LMBV, and 16 (53.3%) samples were negative of LMBV, meaning that 4 (13.3%) samples were tested by qRT-PCR to be positive but by sandwich ELASA negative. Hence, the detection results of sand-wich ELASA showed comparative consistency with that of qRT-PCR, with slightly less sensitive (13.3 % lower) (Table 2).

Page 8: “As shown in Figure 5A&B, the sandwich ELASA could detect as low as 1.25×102 TCID50 or cells?/mL LMBV infected cells lysates” has been revised to “As shown in Figure 5A&B, the sandwich ELASA could detect as low as 1.25×102 cells/mL LMBV infected cells lysates”.

Reviewer 2 Report

This paper is devoted to the development of a fast and relatively simple method for testing a Largemouth bass virus (LMBV) that is a major viral pathogen in largemouth bass culture. Given the high percentage of death of infected fish, the work is very relevant. Of particular interest is the fact that in a modified version of the ELISA method (namely sandwich ELASA), not classical antibodies are used, but aptamers. Aptamers have certain advantages in comparison. with antibodies. Due to the high affinity and specificity of aptamers after many rounds of selections, in addition to their superior properties regarding shelf life, the ability to restore the 3-D structure after denaturation, low molecular size, low immunotoxicity, short time of development and more importantly the wide range of targets, aptamers can replace antibodies anytime in the foreseeable future.

Comments and Suggestions for Authors:

  1. Section5. Assembly of sandwich ELASA.

“Biotin-labeled library as control”.

The sentence is incomplete, or words are missing in it.

  1. Section 2.5. Assembly of sandwich ELASA.

“Subsequently, the wells were washed, and then added to 100 μL streptavidin‐labelled horseradish peroxidase…”

It's better to write like this: Subsequently, the wells were washed, and then 100 µl of horseradish peroxidase labeled with streptavidin was added.

  1. Section 2.9. Detection of LMBV in infected fish tissues by sandwich ELASA.

“RNA was extracted using the SV Total RNA Isolation Kit (Promega) and transcribed by the ReverTra Ace qPCR RT Kit…”.

It is unclear what is meant. How can RNA be transcribed? Probably, it means cDNA synthesis?

  1. Section 3. Results

Figure 3. “Specificity of the sandwich ELASA. Eight controls were used: Control-1, LA38s-SGIV-LA13s sandwich model…”. Control-2, LA38s-FV3-LA13s sandwich model. Control-3, LA38s-EPC cells-LA13s sandwich mode

Perhaps it is worth removing the word "control" along the X-axis and leaving only its number, it is better to give a decoding of specific control samples not in the name of the figure, but directly in the figure, replacing "control 1", etc. by the composition of the sample.

  1. Figure 5. “The SGIV-infected GS cells and uninfected EPC cells as the controls”.

Obviously, there is a missing verb here?

  1. Obviously, the proposed method is faster and does not require special sample preparation and special equipment. But can the demonstrated sensitivity of the method be considered sufficient for mass testing for fish infection? Won't 13,3 % of unidentified infected fish become a source of infection for the rest of the fish population?

Author Response

Response to Reviewer 2 Comments

This paper is devoted to the development of a fast and relatively simple method for testing a Largemouth bass virus (LMBV) that is a major viral pathogen in largemouth bass culture. Given the high percentage of death of infected fish, the work is very relevant. Of particular interest is the fact that in a modified version of the ELISA method (namely sandwich ELASA), not classical antibodies are used, but aptamers. Aptamers have certain advantages in comparison. with antibodies. Due to the high affinity and specificity of aptamers after many rounds of selections, in addition to their superior properties regarding shelf life, the ability to restore the 3-D structure after denaturation, low molecular size, low immunotoxicity, short time of development and more importantly the wide range of targets, aptamers can replace antibodies anytime in the foreseeable future.

Point 1: Section5. Assembly of sandwich ELASA. “Biotin-labeled library as control”. The sentence is incomplete, or words are missing in it.

Response 1: Thanks for the reviewer’s comments. “Biotin-labeled library as control” has been revised to “Biotin-labeled library was used as control”.

Point 2: Section 2.5. Assembly of sandwich ELASA. “Subsequently, the wells were washed, and then added to 100 μL streptavidin‐labelled horseradish peroxidase…”It's better to write like this: Subsequently, the wells were washed, and then 100 µl of horseradish peroxidase labeled with streptavidin was added.

Response 2:  Thanks for the reviewer’s comments. “Subsequently, the wells were washed, and then added to 100 μL streptavidin‐labelled horseradish peroxidase…” has been revised to “Subsequently, the wells were washed, and then 100 µl of horseradish peroxidase labeled with streptavidin was added.”

Point 3: Section 2.9. Detection of LMBV in infected fish tissues by sandwich ELASA. “RNA was extracted using the SV Total RNA Isolation Kit (Promega) and transcribed by the ReverTra Ace qPCR RT Kit…”. It is unclear what is meant. How can RNA be transcribed? Probably, it means cDNA synthesis?

Response 3:  Thanks for the reviewer’s comments. After we extracted RNA, we used the ReverTra Ace qPCR RT kit to perform reverse transcription for cDNA synthesis, according to the kit's instructions. “RNA was extracted using the SV Total RNA Isolation Kit (Promega) and transcribed by the ReverTra Ace qPCR RT Kit…” has been revised as “RNA was extracted using the SV Total RNA Isolation Kit (Promega) and used for reverse transcription by the ReverTra Ace qPCR RT Kit, according to the following reaction procedure”.

Point 4: Section 3. Results Figure 3. “Specificity of the sandwich ELASA. Eight controls were used: Control-1, LA38s-SGIV-LA13s sandwich model…”. Control-2, LA38s-FV3-LA13s sandwich model. Control-3, LA38s-EPC cells-LA13s sandwich mode. Perhaps it is worth removing the word “control” along the X-axis and leaving only its number, it is better to give a decoding of specific control samples not in the name of the figure, but directly in the figure, replacing "control 1", etc. by the composition of the sample.

Response 4: Thanks for the reviewer’s comments. Thanks for the reviewer’s comments. We have revised the names along the X-axis as follows. Apt-SGIV, Apt-FV3, Apt-EPC-lysate, Library-LMBV, Library-EPC-LMBV-lysate, Library-SGIV, Library-FV3 and Library-EPC-lysate replace control 1, control 2, control 3, control 4, control 5, control 6, control 7 and control 8, respectively.

Figure 3. Specificity of the sandwich ELASA. The names along the X-axis are illustrated in detail as follows: Apt-LMBV indicates LA38s-LMBV-LA13s sandwich model. Apt-EPC-LMBV-lysate indi-cates LA38s-the lysates of LMBV-infected EPC -LA13s sandwich model. Apt-SGIV indicates LA38s-SGIV-LA13s sandwich model. Apt-FV3 indicates LA38s-FV3-LA13s sandwich model. Apt-EPC-lysate indicates LA38s-EPC-LA13s sandwich model. Lib-LMBV indicates Library- LMBV-Library sandwich model. Lib-EPC-LMBV-lysate indicates Library- the lysates of LMBV-infected EPC. Lib-SGIV indicates Library- SGIV-Library sandwich model. Lib-FV3 indicates Library- FV3-Library sandwich model. Lib-EPC-lysate indicates Library- EPC -Library sandwich model. SGIV, FV3, EPC and Library were used as control.

Point 5: Figure 5. “The SGIV-infected GS cells and uninfected EPC cells as the controls”. Obviously, there is a missing verb here?

Response 5: Thanks for the reviewer’s comments. “The SGIV-infected GS cells and uninfected EPC cells as the controls” has been revised to “The SGIV-infected GS cells and uninfected EPC cells were used as control”.

Point 6: Obviously, the proposed method is faster and does not require special sample preparation and special equipment. But can the demonstrated sensitivity of the method be considered sufficient for mass testing for fish infection? Won't 13,3 % of unidentified infected fish become a source of infection for the rest of the fish population?

Response 6: Thanks for the reviewer’s comments. The comprehensive assessment of the sandwich ELASA and qRT-PCR should include a range of factors, such as sensitivity, specificity, time range, and operation complexity. In this study, the present results show that the sensitivity of the sandwich ELASA is slightly lower than that of the qRT-PCR. However, qRT-PCR is a laboratory-based method, and usually requires experimental platforms and professional operators. While, the sandwich ELASA can be performed by simple procedure, take short time, and do not require the special operator and equipment, which are key factors in the application of diagnostic method in field. On-site rapid detection of pathogens plays important roles in taking steps in time to control disease. Therefore, the sandwich ELASA is suitable for application in fields. Besides, the sensitivity of the sandwich ELASA could be improved by continually optimizing detection system, which is the priority in our further study.
